# Remdesivir improves biomarkers associated with disease severity in COVID-19 patients treated in an outpatient setting

David Z. Pan[1,2], Pamela M. Odorizzi[1,2], Andre Schoenichen[1], Mazin Abdelghany[1], Shuguang Chen[1], Anu Osinusi[1], Scott D. Patterson [1], Bryan Downie[1], Kavita Juneja[1] & Jeffrey J. Wallin [1✉]

## Abstract

**Background** Remdesivir (RDV) is an intravenous antiviral with activity against SARS-CoV-2 for treatment of hospitalized COVID-19 patients with moderate-to-severe disease. Biomarkers associated with clinical outcomes have been identified for COVID-19, but few evaluated in context of antiviral treatment. Here, we assessed baseline (day 1, prior to first RDV dose) biomarkers and the impact of RDV treatment on longitudinal biomarker readouts.

**Methods** Recently, RDV was evaluated in high-risk, non-hospitalized patients with confirmed SARS-CoV-2 infection and was highly effective at preventing disease progression. The randomized, double-blind, placebo-controlled Phase 3 study included 562 participants who received at least 1 dose of study drug, of which 312 consented for longitudinal biomarker assessments at baseline, day 3, and day 14. We assessed sixteen baseline biomarkers and the impact of RDV treatment on longitudinal biomarker readouts.

**Results** Six well-known, inflammation-associated biomarkers are elevated at baseline in participants meeting the primary endpoint of hospitalization or death by day 28. Moreover, in comparison to placebo, biomarkers in RDV-treated participants show accelerated improvement, including reduction of soluble angiopoietin-2, D-dimer, and neutrophil-to-lymphocyte ratio, as well as an increase in lymphocyte counts.

**Conclusions** Overall, the findings in this study suggest that RDV treatment may accelerate the improvement of multiple biomarkers of COVID-19 severity, which are associated with better clinical outcomes during infection. These findings have implications for better understanding the activity of antiviral treatments in COVID-19.

## Plain language summary

Certain cells and proteins in the blood can act as markers of COVID-19 severity. However, little is known about the impact of antiviral treatments on these markers. Here, we measured protein and cell markers in patient samples before treatment and those taken during the course of COVID-19 in high-risk non-hospitalized patients treated with or without the antiviral remdesivir (RDV). Several markers were improved with RDV treatment, including those associated with normal responses from the immune system and factors involved in blood clotting. These findings further our understanding of the activity of antivirals in COVID-19 and inform future studies to understand how patients with an increased risk of COVID-19 disease progression respond to these treatments.

[1] Gilead Sciences Inc., Foster City, CA 94404, USA. [2]These authors contributed equally: David Z. Pan, Pamela M. Odorizzi. ✉email: Jeff.Wallin@gilead.com

Coronavirus 2019 disease (COVID-19) was first identified in December 2019 and rapidly progressed into a global pandemic that continues to be a major public health concern worldwide. COVID-19 is caused by severe acute respiratory syndrome coronavirus 2 (SARS-CoV-2), a single-stranded RNA virus belonging to the Coronaviridae family[1]. In an analysis of more than 1.3 million laboratory-confirmed SARS-CoV-2 infections that were reported in the United States between January and May 2020, 14% of patients required hospitalization, 2% were admitted to the intensive care unit, and 5% died[2]. The probability of severe COVID-19 is higher in those with pre-existing medical conditions and people ≥60 years old. Men have also been shown to be at higher risk than women for severe outcomes and death, independent of age[3].

Laboratory biomarkers have been identified during the COVID-19 pandemic to inform risk of disease progression and provide information about disease pathogenesis. Early studies described vascular pathology underlying organ damage in severely ill patients, which is associated with higher levels of inflammatory factors, complement activation, and pro-inflammatory mediators in circulation[4]. Hematologic biomarkers that correlate with COVID-19 severity have also been identified; for example, lymphopenia has been consistently associated with worse clinical outcomes[5].

Remdesivir (RDV) is a nucleotide prodrug that binds to the viral RNA-dependent RNA polymerase and inhibits SARS-CoV-2 viral replication through premature termination of RNA transcription[6,7]. RDV has demonstrated clinical benefit in COVID-19 patients hospitalized with moderate-to-severe disease[8,9]. A recent phase 3 clinical trial in non-hospitalized patients with COVID-19 who were at high risk of severe disease further demonstrated that a 3-day course of RDV resulted in an 87% lower risk of hospitalization or death compared to placebo (PINETREE study)[10]. In this study, we investigate biomarkers associated with clinical outcomes for COVID-19 and the impact of RDV treatment on longitudinal biomarker readouts. Overall, the findings suggest that RDV treatment may accelerate the improvement of multiple biomarkers of COVID-19 severity, which are associated with better clinical outcomes during infection.

## Methods
**Study design**. The PINETREE study was a randomized, double-blind, placebo-controlled Phase 3 clinical trial conducted at sixty-four outpatient centers and skilled nursing facilities[10]. The trial was approved by the institutional review board or ethics committee at each site and was conducted in compliance with the Declaration of Helsinki Good Clinical Practice guidelines and local regulations. The names of the IRBs who approved the study are available in Supplementary Data 1. Enrollment included participants with a positive SARS-CoV-2 test at high risk for severe COVID-19, including those ≥ 60 years or ≥ 18 years (or, where permitted by law, 12–17 years weighing ≥ 40 kg) and/or those with one or more of the following risk factors: chronic lung disease, hypertension, cardiovascular disease, diabetes, obesity, immunocompromised, chronic kidney disease, chronic liver disease, cancer, or sickle-cell disease. Individuals vaccinated for COVID-19 could not participate in the study. This trial was conducted according to protocol without substantial deviations and registered with ClinicalTrials.gov (NCT04501952).

**Biomarker sample collection and analysis**. Serum and plasma samples were obtained at baseline, day 3 (on-treatment) and day 14 (post-treatment) in 312 participants of the PINETREE study that consented for optional evaluation of COVID-19 biomarkers.

Biomarker analyses included the following: soluble Angiopoietin 2 (sAng2), interleukin (IL)-6, ferritin, lactate dehydrogenase (LDH), C-reactive protein (CRP), procalcitonin (PCT), activated partial thromboplastin time (aPTT), International Normalized Ratio (INR), prothrombin time (PT), D-Dimer, fibrinogen, eosinophil counts, lymphocyte counts, monocyte counts, and neutrophils counts. All biomarkers were measured at Covance Laboratory Services (Labcorp) using standard laboratory assays and Labcorp provided normal ranges for ferritin, LDH, CRP, aPTT, PT, fibrinogen, eosinophil counts, lymphocyte counts, monocyte counts, and neutrophils counts[11,12]. Analysis of biomarker data was prespecified and conducted at study completion.

**Statistical analysis**. To determine if there were significant changes in biomarker values between baseline (day 1) and post-treatment time points (day 3 or 14) we used a linear mixed effects model (LMM) including age and sex as fixed effect covariates (Supplementary Fig. 1) and patient identification number as a random effect. Each biomarker was tested in its own individual model and any patients with missing data for a given biomarker was removed. All p-values were corrected for multiple testing using Benjamini-Hochberg correction (false discovery rate (FDR)). All source data necessary for the figures in the manuscript are provided in Supplementary Data 1.

**Reporting summary**. Further information on research design is available in the Nature Portfolio Reporting Summary linked to this article

## Results
**Study design**. The study included 562 participants who received at least 1 dose of study drug, of which 312 consented for longitudinal biomarker assessments at baseline (day 1, prior to first RDV dose), day 3, and day 14 (Fig. 1a). Among those who consented for biomarker evaluation, demographics, baseline characteristics, and comorbidities were well-balanced and comparable to those for the larger study (Supplementary Table 1 and Supplementary Fig. 1A)[10]. Biomarker testing was performed on all collection timepoints where enough sample volume was available.

**Comparison of biomarkers at baseline in RDV- and placebo-treated patients**. The impact of RDV on well-established inflammation, coagulation, and hematologic biomarkers was evaluated in longitudinal blood samples at baseline and post-treatment (day 3 or 14) timepoints (Fig. 1a and Supplementary Fig. 3A). Since only a small number of participants in the biomarker subpopulation met the primary endpoint of COVID-19-related hospitalization or death by day 28 (RDV: $n = 2$, Placebo: $n = 4$), biomarker values at baseline were evaluated using a Wilcoxon rank-sum test comparing patients who met the primary endpoint to those who did not. Six well-established biomarkers associated with inflammation were significantly elevated (FDR < 0.05) in patients who met the primary endpoint (Fig. 1b). Fibrinogen was also nominally elevated ($p < 0.05$) in those who met the primary endpoint, while eosinophil counts were nominally decreased ($p < 0.05$; Supplementary Fig. 2). Collectively, these findings suggest that several inflammatory biomarkers at baseline were associated with more severe COVID-19 outcomes in the PINETREE study.

**Biomarkers at on-treatment timepoints compared to baseline**. We next evaluated the impact of RDV treatment on longitudinal biomarker readouts. Elevated levels of C-reactive protein (CRP) have been associated with more severe COVID-19[13–15]. In line with these previous findings, CRP was measured above the

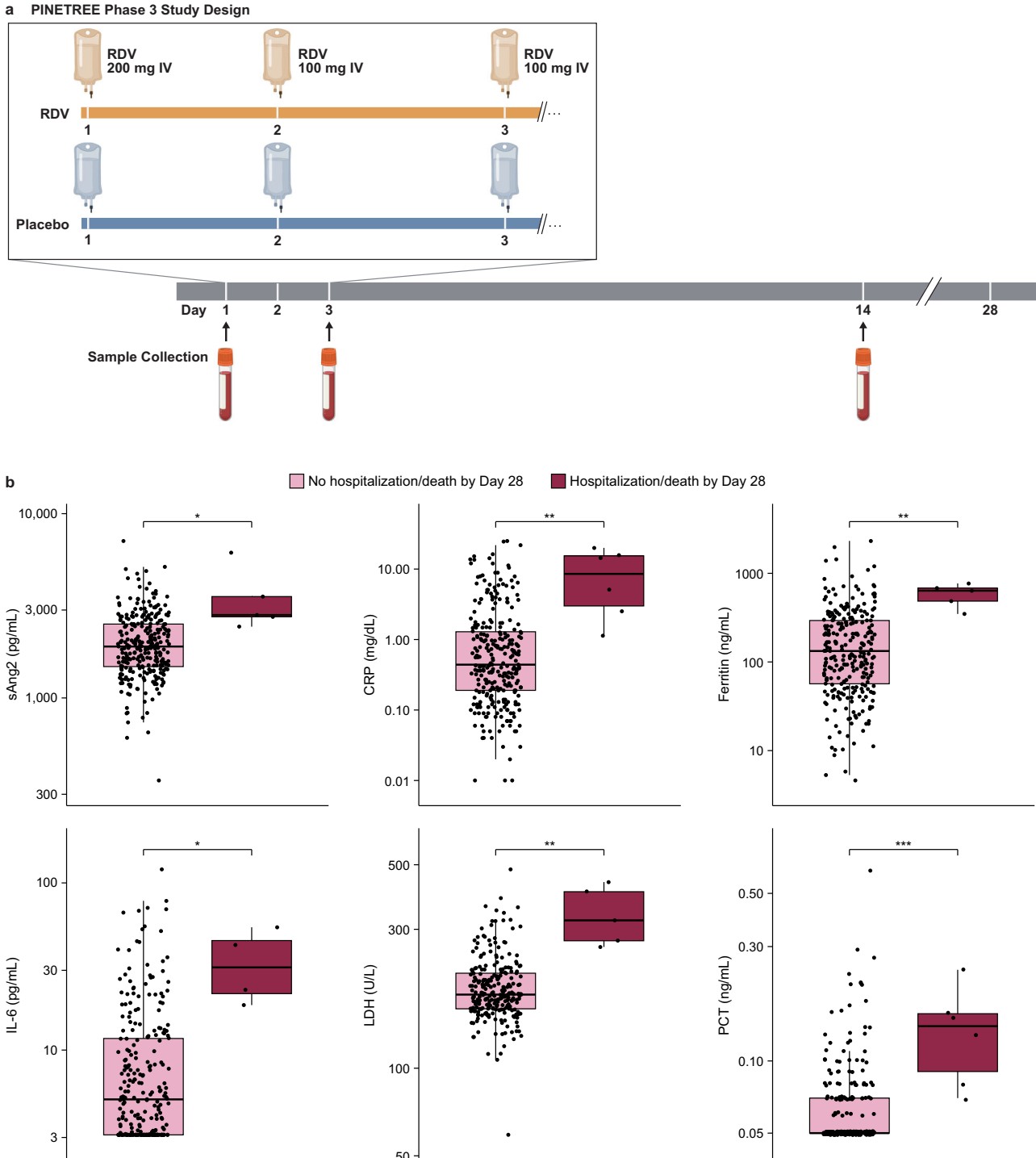

**Fig. 1 Baseline inflammation biomarkers associated with the primary endpoint of the PINETREE study. a** Design of the PINETREE study, a randomized, double-blind, placebo-controlled Phase 3 clinical trial in patients at high-risk of severe COVID-19. Participants were randomly assigned (1:1) to receive remdesivir (RDV) or placebo every day for 3 doses. Serum and plasma were collected for biomarker evaluation in a subset of participants at day 1, day 3, and day 14. **b** Box plots of significantly different (FDR < 0.05) biomarkers at baseline (day 1) in patients who met the primary endpoint of hospitalization or death by day 28 (purple, soluble Angiopoietin 2 (sAng2): $n = 288$; ferritin: $n = 275$; interleukin-6 (IL-6): $n = 267$; lactate dehydrogenase (LDH): $n = 246$; C-reactive protein (CRP): $n = 289$; procalcitonin (PCT): $n = 288$) vs. those who did not (pink; IL-6: $n = 4$; sAng2, ferritin, LDH: $n = 5$; CRP, PCT: $n = 6$). *FDR < 0.05; **FDR < 0.01; ***FDR < 0.001, ****FDR < 0.0001. The line within each box denotes the median and each box extends to the 25th and 75th percentiles. The whiskers indicate 1.5 interquartile range. Significance was determined using a Wilcoxon Rank Sum Test.

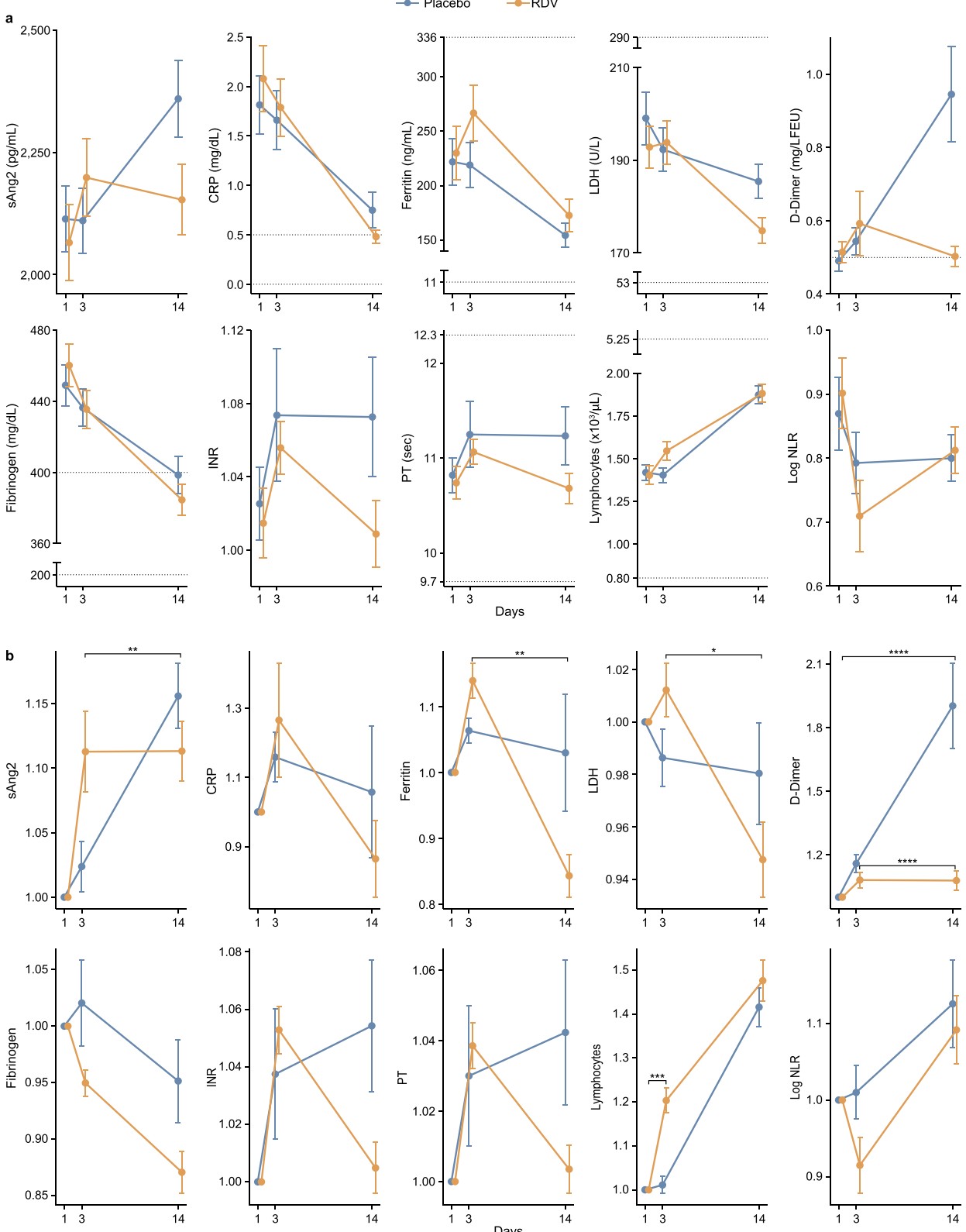

**Fig. 2 Longitudinal changes in inflammation, coagulation, and hematologic biomarkers in remdesivir- vs. placebo-treated patients.** Linear mixed effects model (LMM) used to test differences between day 3 and baseline (day 1), day 14 and day 3, and day 14 and baseline between remdesivir (RDV)- and placebo-treated patients. **a** Longitudinal plots of biomarker absolute value changes (mean with standard error) in RDV- (yellow) vs. placebo-treated (blue) patients. Dashed lines indicate normal ranges or expected values in healthy individuals. **b** Longitudinal plots of fold change (mean with standard error) in biomarkers in RDV- (yellow) vs placebo-treated (blue) patients. *FDR < 0.05, **FDR < 0.01, ***FDR < 0.001, ****FDR < 0.0001. Significance is determined by the LMM. The number of patients used in the LMM for each biomarker is provided Supplementary Data 1.

normal range at baseline, however, there was no significant difference in any biomarkers between treatment arms at baseline (Supplementary Fig. 3A). Unique patterns in biomarker values were observed over time with RDV treatment, as demonstrated by changes in absolute biomarker concentration (Fig. 2a and Supplementary Fig. 3B) and fold change from baseline (Fig. 2b and Supplementary Fig. 3C). A common trend observed was a decrease in biomarker values in RDV- compared to placebo-treated patients by day 14. We observed such a pattern for soluble angiopoietin 2 (sAng2, FDR < 0.05), ferritin (FDR < 0.05), lactate dehydrogenase (LDH, FDR < 0.05), D-dimer (FDR < 0.05), international normalized ratio (INR, $p < 0.05$), and prothrombin time (PT, $p < 0.05$) (Fig. 2 and Supplementary Data 1). These differences between RDV- and placebo-treated patients were most prominent for D-dimer at day 14 compared to baseline (FDR < 0.05, Supplementary Data 1) with continued increase in D-dimer in placebo-treated patients while RDV-treated patients returned quickly to baseline levels. While no biomarkers were found to be significantly different at day 3 compared to baseline in RDV- vs. placebo-treated patients after FDR correction, sAng2 was nominally elevated in RDV- compared to placebo-treated patients ($p = 0.023$; Fig. 2 and Supplementary Fig. 2). An inverse observation was made for lymphocyte count, which was higher at both day 3 compared to baseline (FDR < 0.05) and day 14 compared to day 3 ($p = 0.03$) in RDV- vs. placebo-treated patients (Fig. 2 and Supplementary Data 1).

## Discussion

With direct inhibition of the SARS-CoV-2 polymerase, RDV treatment should reduce viral load and therefore also the corresponding inflammatory response to infection[16]. This may result in concurrent improvement of inflammation, coagulation, and hematologic biomarkers associated with progression to more severe COVID-19. Our analysis demonstrated that RDV treatment may lead to more rapid normalization of several biomarkers (to either normal range or baseline values) associated with COVID-19 severity by day 14, including sAng2, ferritin, LDH, D-dimer, INR, PT, and lymphocyte count. The most prominent differences between RDV- and placebo-treated patients were observed in D-dimer, which was markedly elevated at day 14 in placebo-treated patients in contrast to reversion to baseline in RDV-treated patients, and lymphocyte count, which was elevated in RDV- versus placebo-treated patients at day 3. Previous studies have shown that progression of COVID-19 severity to intensive care unit admission may be predicted by lymphopenia and/or elevated D-dimer levels upon hospital admission[13,17].

Our analysis is limited by the number of patients with biomarker data reaching the primary endpoint. The reduced numbers and the fact that there were only six patients who had a hospitalization or death by day 28 led us to perform a simple rank-based statistical analysis instead of a prognostic modeling approach. Analysis of baseline samples suggests that sAng2, CRP, ferritin, IL−6, LDH, and procalcitonin (PCT) were elevated in study participants with worse outcomes independent of treatment assignment, which is consistent with previously reported associations of baseline inflammatory biomarkers and COVID-19 severity[18–20]. Continued analysis of these biomarkers, as well as additional biomarkers of COVID-19 progression, in the same high-risk outpatient population will be needed to confirm these observations in larger cohorts of patients. Future studies may help further our understanding of the predictive value of these biomarkers, which could prove invaluable in identifying outpatients at increased risk of disease progression who may benefit most from early treatment. Overall, the findings in this study suggest that RDV treatment may accelerate the improvement of multiple biomarkers of COVID-19 severity, which are associated with better clinical outcomes during infection.

## Data availability

Any requests for use of the data in this study must be sent to the corresponding author. The request will be examined with respect to the Informed Consent Form relevant for this clinical trial. Source data for the figures are available in Supplementary Data 1.

## Code availability

All statistical analyses performed in this study were performed using R v4.0.5. No modifications were made to R or the additional packages used for statistical analyses. Any requests for code related to the study should be directed to the corresponding author.

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

## Acknowledgements

We thank the patients and their families, study investigators, and their staff.

## Author contributions

Conception and design: A.O., M.A., K.J., and J.J.W.; methodology: P.M.O., A.S., and J.J.W.; analysis and interpretation of data: D.Z.P., P.M.O., A.S., S.D.P., S.C., B.D., and J.J.W.; writing, reviewing, and editing: D.Z.P., P.M.O., A.S., M.A., A.O., S.D.P., B.D., K.J., and J.J.W.

## Competing interests

All authors are employees of Gilead Sciences and owners of company stock. The trial was designed and conducted by the sponsor (Gilead Sciences) in collaboration with the principal investigators and in accordance with the protocol and amendments. The sponsor collected data, monitored trial conduct, and performed the statistical analyses.
