## [Peer Review File · Communications Medicine]

Reviewers' comments:

Reviewer #1 (Remarks to the Author):

This is a really interesting submission by Pan et al. This study builds upon a recent phase 3 clinical trial (PINETREE Study) demonstrating that COVID-19 patients treated with Remdesivir in the outpatient setting, lowered the risk of hospitalization or death compared to placebo. In this manuscript, longitudinal biomarker analysis was performed on samples from patients from the PINETREE study, and demonstrates that Remdesivir improves COVID-19 disease severity biomarkers (including: circulating sAng2, ferritin, LDH and D-dimer).

All statistical analysis and data presentation are performed to a high standard. This manuscript is well written, however, stylistically does it need to be split into Introduction, Methods, Results and discussion?

Minor comments:

Could the authors please indicate the number of patients, biomarkers measured and timepoints within the abstract.

Could the authors please define baseline. i.e was it day 1 of COVID-19 positivity or day of study enrolment? Do the authors have data to show the median time from positivity to enrolment (i.e. does this differ between the 2x treatment arms?).

Reviewer #2 (Remarks to the Author):

The authors have presented an analysis of the PINETREE trial that is interesting and relevant. I have some comments below, but feel that the manuscript would be relevant to clinicians managing patients with COVID-19

The authors say that the day 1 blood sample is baseline. This is at the start of infection / presentation so would not be the participants normal baseline and I think another form of words would be more suitable.

As only 6 participants reached the primary endpoint, the analysis of these cases and biomarkers needs to be taken cautiously due to the low numbers

The PINETREE manuscript reports the high levels of diabetes / obesity and other co-morbidities in the trial, and I think it would be good to have that data for this sub analysis as well (could be added to supplemental table 1). The effect of these comorbidities on the change in biomarkers might also be relevant, esp as they can be associated with higher baseline inflammatory markers

Presenting both prothombin time and INR seems needless duplication given their direct relationship

The statement that inhibition of viral replication should reduce inflammatory response either referencing to back it up

Responses to Editor/Reviewer comments

We would like to thank the editors and reviewers for their helpful comments and suggestions which we believe have been addressed in the revised manuscript. Edits to the text of the manuscript pertaining to editor and reviewer comments have been highlighted in yellow.

Reviewer #1 (our answers in red):

1. Could the authors please indicate the number of patients, biomarkers measured and timepoints within the abstract. **Thank you for this suggestion. This information has been added to the abstract.**
2. Could the authors please define baseline. i.e was it day 1 of COVID-19 positivity or day of study enrolment? Please **see comment above. Baseline (day 1, prior to treatment first dose) is now defined at the beginning of the manuscript and in the abstract.**
3. Do the authors have data to show the median time from positivity to enrolment (i.e. does this differ between the 2x treatment arms? **Thanks for this suggestion. We have added this information to Supplementary Table 1 and referenced the clinical manuscript for this study.**

Reviewer #2 (our answers in red):

1. The authors say that the day 1 blood sample is baseline. This is at the start of infection / presentation so would not be the participants normal baseline and I think another form of words would be more suitable. **See comment above. Baseline is now defined at the beginning of the manuscript and in the abstract.**
2. As only 6 participants reached the primary endpoint, the analysis of these cases and biomarkers needs to be taken cautiously due to the low numbers. **We agree. Please see our response to second comment by the editor above.**
3. The PINETREE manuscript reports the high levels of diabetes / obesity and other co-morbidities in the trial, and I think it would be good to have that data for this sub analysis as well (could be added to supplemental table 1). The effect of these comorbidities on the change in biomarkers might also be relevant, esp as they can be associated with higher baseline inflammatory markers. **We thank the reviewer for the suggestion. We have added the information for the comorbidities into Supplementary Table 1. As there are no significant differences between the remdesivir and placebo arms (chi-squared test, $p > 0.05$), we do not expect these to be the cause any differences in the baseline biomarkers levels.**
4. Presenting both prothrombin time and INR seems needless duplication given their direct relationship. **Although we agree that these are similar, we did find subtle differences in Figure 2a (absolute value of biomarker, mean with standard error) and for completeness would like to leave data for both markers in.**
5. The statement that inhibition of viral replication should reduce inflammatory response either referencing to back it up. **Thank you for pointing this out. A reference has been added.**